# Longitudinal Comparison of Neutralizing Antibody Responses to COVID-19 mRNA Vaccines after Second and Third Doses

**DOI:** 10.3390/vaccines10091459

**Published:** 2022-09-03

**Authors:** Alexa J. Roeder, Megan A. Koehler, Paniz Jasbi, Davis McKechnie, John Vanderhoof, Baylee A. Edwards, Maria J. Gonzalez-Moa, Alim Seit-Nebi, Sergei A. Svarovsky, Douglas F. Lake

**Affiliations:** 1School of Life Sciences, Arizona State University, Tempe, AZ 85281, USA; 2College of Health Solutions, Arizona State University, Phoenix, AZ 85004, USA; 3School of Molecular Sciences, Arizona State University, Tempe, AZ 85281, USA; 4Sapphire/AXIM Biotechnologies, San Diego, CA 92121, USA

**Keywords:** mRNA vaccines, SARS-CoV-2, neutralizing antibody, lateral flow assay

## Abstract

COVID-19 mRNA vaccines protect against severe disease and hospitalization. Neutralizing antibodies (NAbs) are a first-line defense mechanism, but protective NAb responses are variable. Currently, NAb testing is not widely available. This study employed a lateral flow assay for monitoring NAb levels postvaccination and natural infection, using a finger-stick drop of blood. We report longitudinal NAb data from BNT162b2 (Pfizer) and mRNA-1273 (Moderna) recipients after second and third doses. Results demonstrate a third dose of mRNA vaccine elicits higher and more durable NAb titers than the second dose, independent of manufacturer, sex, and age. Our analyses also revealed that vaccinated individuals could be categorized as strong, moderate, and poorly neutralizing responders. After the second dose, 34% of subjects were classified as strong responders, compared to 79% after the third dose. The final months of this study coincided with the emergence of the SARS-CoV-2 Omicron variant and symptomatic breakthrough infections within our study population. Lastly, we show that NAb levels sufficient for protection from symptomatic infection with early SARS-CoV-2 variants were not protective against Omicron infection and disease. This work highlights the need for accessible vaccine response monitoring for use in healthcare, such that individuals, particularly those in vulnerable populations, can make informed vaccination decisions.

## 1. Introduction

COVID-19 mRNA vaccines BNT162b2 (Pfizer) and mRNA-1273 (Moderna) dramatically reduced the incidence of severe disease and hospitalization due to SARS-CoV-2 infection [1,2]. One goal of all COVID-19 vaccines is to induce neutralizing antibodies (NAbs), which prevent the virus from infecting host cells [3,4,5]. Vaccines also induce antiviral T cells that kill infected cells, but when cytotoxic T cells are engaged, the host is already infected and virus is actively replicating, creating the potential for transmission. In individuals vaccinated with the ancestral SARS-CoV-2 spike protein-encoding mRNA, breakthrough infections likely occur due to waning levels of neutralizing antibodies [6,7,8,9,10] or infection with a variant of concern, such as one of the Omicron variants [11,12].

The principle neutralizing domain of SARS-CoV-2 spike glycoprotein is the receptor-binding domain (RBD) [13], although additional NAbs have been observed that target the N-terminal domain (NTD) of spike [14]. Antibodies that target the RBD of spike have potent neutralizing activity [15], but are particularly prone to loss of efficacy as variants arise. RBD has been shown to be highly mutated in several variants of concern (VOCs), most significantly in the Omicron variant and subvariants [16]. The assay employed in this study uses the RBD of SARS-CoV-2 USA-WA1/2020 spike for detection of Nabs that inhibit RBD binding to host the cellular receptor, angiotensin-converting enzyme 2 (ACE2). For the purposes of this study, NAbs that obstruct the RBD-ACE2 interaction are quantitatively measured (Figure 1) using a rapid lateral flow assay (LFA), previously described [17]. The RBD-NAb LFA has been shown to detect vaccine-induced NAbs as well as those induced by natural infection [18].

As the COVID-19 pandemic transitions into an endemic phase with potential surges occurring in different geographic locations at certain times of year, monitoring vaccine-induced immunity may be an important component of healthcare. Specifically, monitoring NAbs may also allow individuals and their healthcare providers to gauge when a booster might be warranted as NAbs wane, especially in vulnerable and immunosuppressed populations. Prevention of SARS-CoV-2 infections could reduce opportunities for the virus to mutate into more infectious or pathogenic VOCs, as has been demonstrated throughout the pandemic [19,20]. Although measuring NAb titers has been suggested as a correlate of protection [21,22,23], it is not known what the risk of infection is, as NAb titers wane after both vaccination and natural infection. In addition to waning titers, it is known that natural infection alone may not elicit measurable levels of NAbs [24], especially when infections are mild [25]. However, breakthrough infections have been reported to boost NAbs in vaccinated individuals [26].

We undertook this study to learn the durability of RBD-NAb titers in previously uninfected COVID-19 vaccine recipients after second and third vaccine doses. Here we report longitudinal second- and third-dose data from 302 healthy individuals and demonstrate the importance of a third dose for RBD-NAb longevity, independent of mRNA vaccine manufacturer. Since Omicron-variant infections became widespread during the final few months of our study, we present breakthrough infection data pre- and post-Omicron surge in the United States, specifically Arizona. This is the first report that employs a rapid NAb test to longitudinally measure levels of neutralizing antibodies after a second and third mRNA vaccine dose.

## 2. Materials and Methods

### 2.1. Experimental Design

The purpose of this study was to quantify RBD-NAb levels prior to and post-mRNA vaccination using a rapid lateral flow assay developed previously in our laboratory [17]. Participants were tested at variable timepoints prior to the second vaccine dose, then monthly for the remainder of their enrollment. Finger-stick blood samples were obtained within one week prior to 2nd dose, and 2–4 weeks post-2nd-dose. Longitudinal data were collected monthly according to the date of participants’ post-2nd-dose test (*n* = 265). As third vaccine doses (boosters) became available, we began monitoring third-dose recipients’ RBD-NAbs in the same manner (*n* = 142). Some of the individuals in the 3rd-dose cohort were newly enrolled, although many continued from prior enrollment in the 2nd-dose study (*n* = 105), such that paired longitudinal data were collected from 105 participants.

### 2.2. Participant Recruitment

Male and female adults ranging in age from 18–80 years old upon entry into the study were recruited and enrolled with informed consent under a protocol approved by the institutional review board at Arizona State University (IRB #0601000548). Participants were enrolled at various times throughout vaccination and returned for monthly testing until the study completed. Reasons for termination of enrollment prior to completion of the study included breakthrough infection and participant dropout.

### 2.3. Exclusion/Inclusion Criteria

Individuals under the age of 18 and older than 80 years old at the time of enrollment were excluded from this study, as well as individuals with PCR-confirmed natural infection prior to vaccination. Individuals with symptomatic and laboratory-confirmed breakthrough infections as well as those that were asymptomatic but detected due to routine testing were excluded from data analyses; however, subjects were not required to test routinely for the purpose of the study. Patients actively undergoing cancer therapies or treatment for severe autoimmune disease with systemic immunosuppressive therapy were also excluded. The study population included healthy individuals between 18 and 80.

### 2.4. Longitudinal NAb Monitoring Using a Rapid RBD-NAb LFA

Longitudinal monitoring was conducted using ten microliters of blood obtained from a finger stick using a 28-gauge, 1.8 mm pressure-activated safety lancet. Blood was transferred to the test cassette by volumetric micro transfer pipette and chased with two drops (60 μL total) of buffer. The test ran undisturbed on a flat surface for ten minutes, then read using a portable densitometer (Detekt RDS-2500) to quantify control and test line densities. Data were exported to Microsoft Excel using the Detekt Data Manager software (Detekt Biomedical, Austin, TX, USA) and recorded as de-identified participants in a master file.

### 2.5. Data Analyses

Raw data were converted to percent neutralization by normalizing values according to the limit of detection previously defined for the rapid test [17]. The equation used to calculate percent neutralization was: 1-(test line density/limit of detection)*100%. Limit of detection of least neutralizing serum was a density of 942,481 density units. Normalized data were subsequently analyzed using GraphPad Prism 9.0 (GraphPad Software, San Diego, CA, USA). Unsupervised hierarchical clustering, principal component analysis (PCA), and significance testing with correction for false discovery rate (FDR) were performed and results were visualized using R version 4.1.2, https://www.R-project.org/. Percent neutralization graphs were made using Microsoft Excel version 16.6.1. Post hoc power analysis was performed and critical *R^2^* visualized using G*Power version 3.1.9.6. Missing values were imputed using a subject-wise *k*-nearest-neighbor algorithm. Conversions between test line density, percent neutralization, and NAb ranges are demonstrated in Appendix A. All normalized or raw de-identified data can be made available upon request to the corresponding author.

## 3. Results

We previously developed a rapid test to measure levels of neutralizing antibodies to SARS-CoV-2 using 10 μL of finger-stick peripheral blood [17]. The test quantitatively measures antibodies that inhibit spike protein RBD-GNS from binding to ACE2 (neutralizing antibodies) in a lateral flow assay. Test line density is quantified in a lateral flow assay reader, and converted to percent neutralization and NAb titers compared to a live-virus focus reduction neutralization test, as previously reported [18]. The rapid test has been shown to measure NAbs induced after a natural infection and after mRNA vaccination [17]. In this study, we monitored 302 individuals’ NAb responses after they received a second and/or third dose of COVID-19 mRNA vaccine. All study participants were healthy with no reported current comorbidities. We evaluated the durability of RBD-NAbs in COVID-19 mRNA vaccine recipients monthly for 6–8 months after the second dose, and for 6–8 months after the third dose. Study participants in whom breakthrough infections occurred were discontinued from the study at the time of the PCR-confirmed SARS-CoV-2 infection.

The study began in December 2020 after participants began to receive their first dose of either BNT162b2 or mRNA-1273. If participants were available, NAb levels were measured within seven days prior to the second and third vaccine doses. At the height of recruitment, 2–4 weeks after the second vaccine dose, 234 participants were enrolled. Nabs were measured 2–4 weeks after the second or third dose, then monthly for 6–8 months. If a participant missed a month, they remained in the study until months 7 or 8 or until the study was completed. The number of vaccinated participants in the second- and third-dose groups and corresponding demographics are shown in Table 1. A flowchart demonstrating sample sizes of the second- and third-dose groups is shown in Appendix A.

### 3.1. Longitudinal NAb Titers

The range of individuals’ NAb titers is shown at ≤1 week prior to a second vaccine dose, 2–4 weeks after dose two, and then monthly for 6–8 months for recipients of either BNT162b2 or mRNA-1273 vaccine (Figure 2A). Two to four weeks after the second dose, mean NAb titers increased from between 1:40 and 1:80 (35% neutralization) to between 1:160 and 1:320 (71% neutralization). However, 28% (65/234) of second-dose recipients’ NAb titers did not reach 50% neutralization 2–4 weeks after the second vaccine dose, when antibody titers are typically at their peak. Between months two and three after a second dose, mean titers declined to between 1:80 and 1:160 (56% mean neutralization), and remained relatively constant between titers of 1:80 and 1:160.

Study participants received their third vaccine dose an average of 7 months after their second dose, when mean titers were between 1:40 and 1:80 (~25% neutralization). Two to four weeks after a third dose, mean NAb titers increased to ≥1:640 (92% neutralization; Figure 2B). In contrast to the second dose of mRNA vaccine in which mean titers declined rapidly, NAbs remained elevated in third-dose vaccine recipients and did not drop below 50% mean neutralization, even at 6–8 months.

### 3.2. Homologous and Heterologous mRNA Vaccination

Our study design allowed for comparison of longitudinal NAb data between homologous booster vaccine recipients (i.e., those who received two or three doses from the same manufacturer) versus heterologous booster vaccine recipients, or those who received two or three doses from different manufacturers (mixed vaccines). After a second dose, recipients of mRNA-1273 elicited significantly stronger NAb responses at the post-second-dose timepoint through month 5 (*p* < 0.05−0.0005; Figure 3A). There were no significant differences in the NAb responses to different vaccines after the third dose (Figure 3B). It should be noted that all but one individual in the BNT162b2 group (*n* = 62) shown in Figure 3B had previously received two BNT162b2 vaccine doses. However, in the mRNA-1273 third-dose group (*n* = 80), 31 individuals received mRNA-1273 previously, while 48 individuals previously received BNT162b2. Further, we found that receiving a third dose of either mRNA vaccine was more important than adherence to one particular manufacturer (i.e., “mix and match”, Figure 3C), although at month 5, we observed a significant increase in percent neutralization of the mixed vaccine population, relative to the group that received three BNT162b2 doses (*p* < 0.05). The difference was resolved at months 6–8 post-third-vaccination, and mean titers remained in the ≥1:160, <1:320 range, independent of vaccine manufacturer or mixed vaccines.

### 3.3. Sex-Based Differences

Sex and age have previously been implicated as factors in COVID-19 vaccine efficacy [26]. Although reports of sex-based differences with respect to humoral immunity elicited by mRNA vaccination have yielded conflicting results, [27,28] antibody responses to other viral vaccines have been shown to be more robust in females [29].

We evaluated NAb levels in male and female vaccine recipients who were <65 and those who were ≥65 after second and third doses. No sex-based differences were observed after second dose (Appendix AA) and we observed only a slight increase in mean titers of females (*p* = 0.05) 2–4 weeks post-third-dose (Appendix AB), although both males and females had means of ≥90% neutralization (NAb titer >1:640), 2–4 weeks post-third-dose.

### 3.4. Age-Based Differences

When evaluating age, differences were observed only after the second dose, and when grouped as ≥65 or <65 years old (Appendix AC). Significant age-based differences were observed at post-second-dose and month 3 timepoints only (*p* < 0.05 and *p* < 0.005, respectively). When grouped at other age ranges, such as above and below 50 years, no differences were observed (data not shown). In vaccine recipients younger than 65, mean titers declined from between 1:320 and 1:160 (76% neutralization, mean + SEM) to between 1:160 and 1:80 (44% neutralization, mean + SEM) at 2 and 6–8 months after the second dose, respectively (Appendix AC). However, by month 3 following the second dose in the 65-and-older cohort, mean titers declined to between 1:80 and 1:40 at 35% neutralization (mean + SEM), compared to 51% neutralization (mean + SEM) in the younger cohort (*p* < 0.005). Significant differences disappeared after month 4. In contrast, for both age groups after the third dose (Appendix AD), mean titers were sustained between 1:160, and 1:320 at 6–8 months, or ≥ 64% neutralization (mean + SEM), further strengthening the value of a third vaccine dose in a population 65 and older.

### 3.5. Sex and Age-Based Differences

When evaluating sex and age together, females < 65 had a slight tendency for higher mean titers compared to females ≥ 65, after both the second and third vaccine doses (Appendix AE). Statistically significant differences were observed when comparing females ≥ 65 and < 65 only at the month 3 timepoint after the second dose (*p* < 0.005), and differences disappeared by month 4. Additionally, only a modest significant difference in mean NAb levels was observed between females ≥ 65 and < 65 (*p* < 0.05), within one week prior to their third dose (Appendix AF). No additional differences were observed at subsequent timepoints following a third vaccine dose when comparing sex and age together.

### 3.6. Classification of Vaccine-Induced NAb Responses Using Unpaired Data

Unpaired samples were analyzed by second and third doses. Unsupervised hierarchical clustering was performed on 265 subjects using percent neutralization data collected pre-second- and post-second-dose, as well as at months 2–8, and the associated dendrogram was analyzed for optimum grouping. As shown in Figure 4A, data were best arranged into two groups. The same analysis was performed using percent neutralization of 142 subjects collected similarly at pre-third- and post-third-dose, and at months 2–8. These data advocated most strongly for classification as two groups as well (Figure 4B).

Subjects were grouped as indicated by unsupervised clustering. Percent neutralization was graphed at each timepoint and then analyzed by dose. Following both the second and third doses, some subjects exhibited a strong neutralization response to vaccination while others exhibited a relatively tempered response in comparison (Figure 4C,D). As such, the former group was labeled vaccine strong responders (VSRs) and the latter termed vaccine moderate responders, or VMRs. After the second dose, 201 subjects were classified as VMRs (~76%), while 64 subjects were classified as VSRs; interestingly, this observation was reversed following the third dose, with 112 subjects (~79%) classified as VSRs by immune response and only 30 subjects classified as VMRs. Percent neutralization of VSR and VMR groups was significantly different at all timepoints for both second (Figure 4C) and third (Figure 4D) doses; both groups showed significant differences in percent neutralization at the *q* = 0.001 level for all timepoints except pre-third-dose, which was significant at *q* = 0.05.

PCA was performed for the second- and third-dose data and respective score plots were analyzed for separation and percent variance (Figure 4E,F). Using percent neutralization data from pre-second-dose to months 6–8, VSR and VMR groups showed appreciable separation and accounted for 63.3% of between-group variance (Figure 4E). Using third-dose data, even greater separation between VSR and VMR groups was noted, while PC1 and PC2 accounted for 70.3% of observed variance (Figure 4F). Cumulatively, PCA results confirm major differences in percent neutralization across time between VSR and VMR groups indicated by unsupervised clustering (Figure 4A,B) and identified by significance testing (Figure 4C,D).

Subjects were also analyzed by dose, irrespective of response group. In Figure 5, percent neutralization in response to second and third doses is graphed for each timepoint from pre-dose to months 6–8. At pre-second and pre-third doses, no significant difference in percent neutralization was observed (*q* > 0.05). However, at post-second and post-third doses, a significant difference in percent neutralization was observed between recipients (*q* < 0.001), a trend that continued through months 6–8.

### 3.7. Classification of Vaccine-Induced NAb Responses using Paired Longitudinal Data

Unsupervised hierarchical clustering on data from 105 subjects from which we obtained concomitant second- and third-dose data across 14 timepoints revealed a tendency for two groups (Figure 6A). Subjects were grouped as indicated by the dendrogram, and percent neutralization across the 14 timepoints with coexistent data was graphed to assess longitudinal vaccine durability between groups. As can be seen in Figure 6B, two distinct trends in percent neutralization emerged when subjects were grouped as indicated by clustering. VSRs (*n* = 42) showed the strongest response to the second dose and remained VSRs after the third dose. The other group, VMRs, to which the majority of subjects with concurrent data were assigned (*n* = 63), showed a moderate response to vaccination in comparison to the VSR group, especially with regard to the second dose, and remained VMRs after the third dose. A Mann–Whitney *U* test was used to assess significant differences in percent neutralization between identified groups across measured timepoints and type I error was controlled for FDR (i.e., *q* values reported). Notably, no significant between-group difference was observed pre-second dose (*q* > 0.05), although both groups were significantly different from post-second dose to months 14–16. VSR and VMR groups showed significant differences in percent neutralization at the *q* = 0.001 level following the second dose as well as at months 2, 4, 5, 11, 12, 13, and 14–16. Groups were also significantly different from each other at month 3, pre-third-dose, and at month 10 (*q* < 0.01), while groups differed least significantly at months 6–8 and post-third-dose (*q* < 0.05*)*.

PCA was performed for the paired longitudinal data between VSR and VMR subjects for which parallel second- and third-dose data were obtained (*n* = 105) and the two-dimensional score plot was analyzed for separation and percent variance (Figure 6C). Using percent neutralization data from pre-second dose to months 14–16, VSR and VMR groups showed appreciable separation and accounted for 48.5% of between-group variance. Cumulatively, PCA results confirm major differences in percent neutralization across time between VSR and VMR groups indicated by unsupervised clustering (Figure 6A) and identified by significance testing (Figure 6B).

A post hoc power analysis was performed to compute achieved β given α (0.05), sample size (*N* = 105), and effect size (*R^2^ =* 0.3). With these parameters, power was calculated for a two-tailed, random-model linear multiple regression (exact test) with 14 predictors; power (i.e., 1−β) was calculated to be 0.974 (Appendix A).

Clustering analyses of paired longitudinal data statistically favored vaccinated subjects categorized as two groups. However, a third, loosely clustered population (*n* = 7) can be visualized within the VMR group (Figure 6C, red circle). Although this minor subgroup is not statistically defined, all seven subjects showed a particularly weak neutralization response to vaccination and subsequent poor durability. As such, this group was identified as vaccine poor responders (VPRs). Average percent neutralization at post-second- and post-third-dose timepoints was 31% and 54%, respectively (Figure 6D). The seven VPRs identified in Figure 6C were an average age of 68 years old and were 57% male.

### 3.8. Breakthrough Infections

During the last 4 months of this study (December 2021–March 2022), Omicron became the dominant variant circulating in the study population. Prior to the Omicron surge in the United States and throughout the Delta variant wave, we observed only 14 PCR-confirmed breakthrough infections in our study population of mRNA vaccine recipients. Ninety-three percent (13/14) of these breakthrough infections occurred when NAb titers were < 1:80 (Appendix A). These thirteen breakthrough infections demonstrated an average of 16% neutralization (median 17%, range 0–84%) prior to infection. Only one individual in the pre-Omicron breakthrough population had a NAb titer >1:80.

Conversely, 14 individuals that had high NAb titers (≥1:640 [n = 9], ≥1:320 [n = 2] and ≥1:160 [n = 3], average 90% neutralization) after receiving a third mRNA vaccine dose became symptomatic with a breakthrough infection, as Omicron had already become the dominant variant in circulation [30], (range 67–99%, median 97%) (Appendix A). The majority of the individuals with Omicron breakthrough infections had titers ≥ 1:320 (≥80% neutralization) to the ancestral-strain RBD used in our test prior to natural infection. Population demographics, vaccination data, and time between Nab and PCR testing are shown in the Appendix A.

## 4. Discussion

In this manuscript, we report durability of RBD-NAb levels elicited by second and third doses of COVID-19 mRNA vaccines, BNT162b2 and mRNA-1273, using a quantitative rapid test. Our study demonstrates three main findings: (i) a third vaccine dose elicits NAb titers that are higher and more durable than a second dose; (ii) the increase in NAb titers and durability of response are independent of vaccine manufacturer; and (iii) high-titer NAbs elicited by a third vaccine dose for which mRNA sequences encode spike glycoprotein of the Wuhan-Hu-1 SARS-CoV-2 isolate do not protect against infection and symptomatic disease with Omicron variants, but appear to protect against severe disease and hospitalization.

One striking feature of our data is the wide range of individual NAb titers elicited by a second vaccine dose. This observation is most notable in BNT162b2 vaccine recipients compared to mRNA-1273 vaccine recipients post-second-dose (Figure 3A). When NAb titers should be at their highest levels, 2–4 weeks after vaccination, nearly one-quarter of BNT162b2 recipients did not reach NAb levels of 50% [18]. Two to four weeks after the second dose, mRNA-1273-vaccinated individuals exhibited significantly higher NAb titers relative to BNT162b2-vaccinated individuals (*p* < 0.0005) and retained significance through month 5 (*p* < 0.05–0.0005). By six months after the second dose, both groups fell below 50% neutralization and no significant difference was observed. These findings demonstrate that although the vaccines elicit high-titer NAbs in the majority of recipients, a large portion of a population that is currently considered to be fully vaccinated with two doses may not have mounted adequate protective NAb responses. The wide range of individual responses highlight the need for accessible NAb testing for individualized vaccine response monitoring. However, we wish to note that this study did not measure T-cell reactivity, an important component of long-term immunity postvaccination [31].

We observed that a third mRNA vaccine dose was highly effective in inducing NAb titers >1:640 at 2–4 weeks postvaccination, independent of vaccine manufacturer (Figure 3B). Both BNT162b2 and mRNA-1273 groups exhibited higher and more durable titers (≤1:320, >1:160) that neutralized ≥60% 6–8 months after vaccination, although mean titers for the mRNA-1273 group were slightly higher. Interestingly, at month 5, the mixed vaccine group demonstrated a significant increase in neutralization relative to the group that received all three BNT162b2 doses (Figure 3C). No differences in mean titers between mixed and non-mixed vaccine groups were observed at 6–8 months; however, when comparing *median* titers of mixed and non-mixed groups, we observed an 8–12% neutralization increase in the group that received three mRNA-1273 doses compared to those that received three doses of BNT162b2 or mixed vaccines, respectively. The group that received all three mRNA-1273 vaccines had mean titers more proximal to the 1:320 to 1:640 range, measured 6–8 months following their last vaccination, suggesting increased durability for mRNA-1273 vaccine recipients.

Statistical analysis of unpaired percent neutralization data revealed two distinct groups within our second- (*n* = 265) and third-dose (*n* = 142) populations (Figure 4), moderate responders (VMRs), and strong responders (VSRs). Analyses of paired second- and third-dose longitudinal data (*n* = 105) also demonstrated VMRs and VSRs as two statistically distinct groups; however, we observed a small subgroup of subjects (*n* = 7) statistically clustered within the VMR group that we called VPRs, shown in Figure 6C. Paired longitudinal data for these seven subjects demonstrated suboptimal RBD neutralization in response to vaccination, combined with poor durability of those responses, rarely neutralizing greater than 50% (Figure 6D). Although a minor subpopulation, VPRs highlight a group of otherwise healthy individuals that fail to mount high-titer protective antibody responses to mRNA vaccination, and likely do not know that their vaccine did not elicit high-titer protective NAbs. Ongoing investigations in our laboratory aim to investigate T-cell differences in VSR, VMR, and VPR groups, but are beyond the scope of this report.

Nabs as a correlate of protection remain undefined and are complicated by evolving variants, demonstrated by data shown in Appendix A. As the technology is available to rapidly measure titers of protective Nabs and is fairly easily modified to the circulating variant, it is possible to deploy variant-specific rapid tests on a large scale to establish a probability of infection based on titers of RBD-Nabs that reflect the variant(s) in circulation.

Limitations of our study include the surrogate nature of the rapid test, which detects NAbs that block RBD from binding to ACE2. The test does not detect antibodies that neutralize by binding to the N-terminal domain [14] or outside the RBD-ACE2 binding site [13,15,32]. However, the RBD is a major determinant of neutralization [33], and FDA-approved therapeutic monoclonal antibodies block RBD from binding to ACE2 [34,35].

Another limitation is in the cohort of mixed vaccines. Study participants who mixed vaccines received BNT162b2 as their first and second doses and mRNA-1273 as a third dose, but we had only one participant who received two doses of mRNA-1273 followed by a third dose of BNT162b2 (Appendix A). Although mean NAb titers declined to lower levels in BNT162b2 vaccine recipients than in mRNA-1273 recipients, they were both in the <1:160, ≥1:80 range at 6–8 months.

It is well-established that COVID-19 mRNA vaccines protect against severe disease and hospitalization, but protection wanes over time [36] and a probability of infection at given NAb titers is not known, although multiple models have been reported [23,37]. For example, is an individual with a NAb titer of 1:320 considered to be ‘twice’ as well-protected from infection as someone with a NAb titer of 1:160? This question is difficult to answer and is confounded by evolving SARS-CoV-2 variants such as Omicron and its subvariants, as well as other effector functions of binding antibodies and cell-mediated immune responses.

Prior to December 2021 and throughout the Delta wave, breakthrough infections occurred almost exclusively in study participants whose NAb titers were <1:80, suggesting that titers below 1:80 might not protect against symptomatic infection. However, when Omicron displaced Delta as the dominant variant in circulation, even individuals with high titers (>1:320) elicited by a third vaccine dose of vaccine became symptomatically infected.

To be considered fully vaccinated at the time of this writing, one must have received two doses of either mRNA vaccine or a single dose of Ad26.CoV2.S [38]. Our data suggest that a third dose provides a more durable NAb response than two doses and support the likely need for subsequent booster vaccines after the second and third doses.

As is evident by surges in Omicron cases across the United States [39], many of which are symptomatic infections in “boosted” individuals months out from their last vaccine [11,39], additional vaccinations containing one or more Omicron variant(s) spike-encoding mRNA may be warranted. It is unclear at this time what the evolutionary space of SARS-CoV-2 will be, and what the frequency of vaccination to protect against symptomatic infection will be. Our results, along with many other reports of breakthrough infections in third-dose recipients [39], implicate the importance of variant-specific booster vaccines. Furthermore, individuals’ responses to variant-specific vaccines could be monitored using a rapid NAb test.

While many have called for NAb test accessibility, we acknowledge that this remains a heavily debated topic, as the implications of that knowledge are not clearly defined. We wish to highlight the importance of quantifying protective antibodies, particularly in people who have high contact with vulnerable populations, which include but are not limited to immunosuppressed and immunodeficient individuals, cancer and transplant patients, and elderly people, in addition to vulnerable individuals themselves. This is the first study that reports individual mRNA vaccine-induced NAb responses longitudinally using a rapid test. It demonstrates that an accessible NAb test may prove useful so that individuals and their healthcare providers can make informed decisions about vaccination and boosting, based on their risk tolerance potential for infection with SARS-CoV-2.

## Figures and Tables

**Figure 1 vaccines-10-01459-f001:**
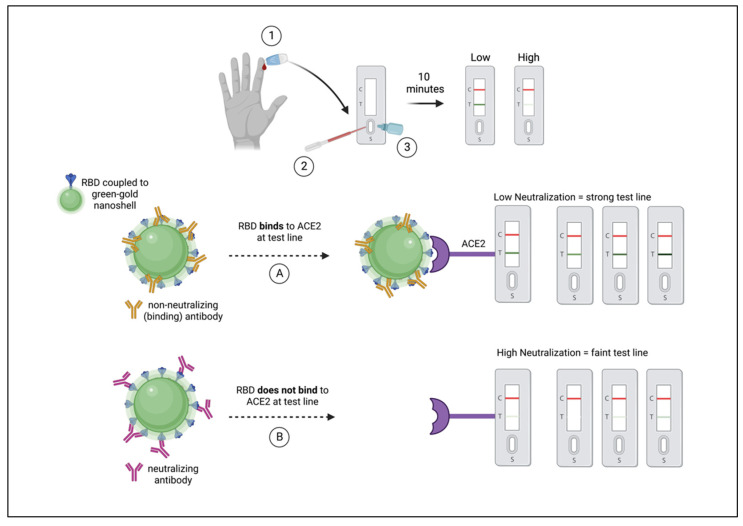
Schematic diagram of NAb LFA principle/mechanism. (**1**) Fingerstick blood is obtained using a pressure-activated safety lancet. (**2**) Ten microliters of blood are transferred to the sample port on a test cassette. (**3**) Buffer is applied to the sample port. (Left to right) RBD of spike (blue) is shown coupled to a green-gold nanoshell (green-GNS). Non-neutralizing antibodies (gold) are shown to bind outside of the RBD, such that in outcome (**A**) RBD-GNS is available to bind ACE2, seen as a strong test line. Neutralizing antibodies (maroon) are shown binding to RBD, obstructing the interaction between antigen and receptor, such that in outcome (**B**) RBD does not bind to ACE2, observed as a faint or absent test line.

**Figure 2 vaccines-10-01459-f002:**
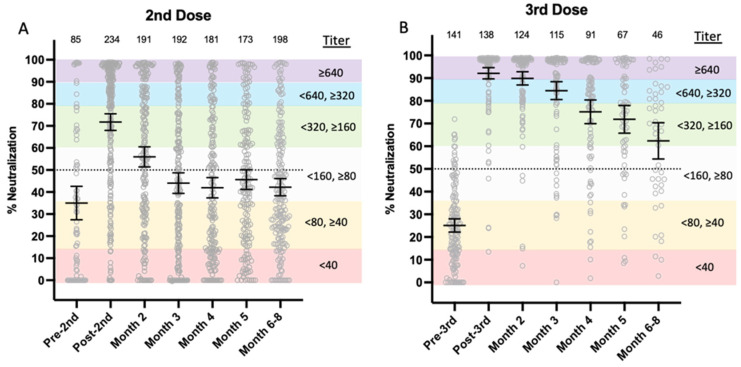
Overall comparison of second- and third-dose COVID-19 mRNA vaccine-induced NAb durability. (**A**) Second dose; (**B**) third dose. Gray circles represent percent neutralization from each study participant vaccinated with either BNT162b2 or mRNA-1273 within one week prior to vaccination (Pre-2nd), 2–4 weeks postvaccination (Post-2nd), then monthly after either second or third doses. The horizontal black lines with error bars represent mean with 95% confidence intervals. Dotted line is 50% neutralization. Numbers above each x-axis time point indicate the number of participants. Reciprocal titer ranges corresponding to % neutralization are shown on the graphs as shaded purple (≥1:640), blue (<1:640, ≥1:320), green (<1:320, >1:160), light grey (<1:160, ≥1:80), orange (<1:80, ≥1:40), and red (<1:40) as reported in [18]. Percent neutralization was calculated as 1-(test line density/limit of detection) x100% (limit of detection test line density = 942,481), as detailed previously [18].

**Figure 3 vaccines-10-01459-f003:**
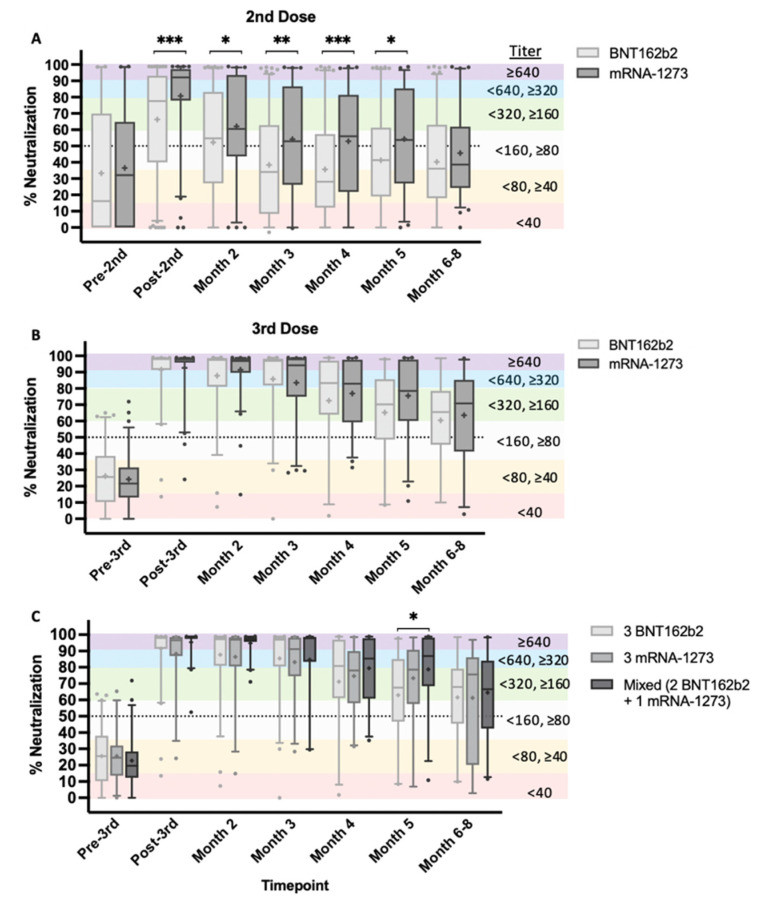
NAb durability of second and third mRNA vaccine doses by manufacturer. NAb test timepoints represented as percent neutralization are pre-2nd/3rd (within one week), post-2nd/3rd (2–4 weeks after second or third dose), and monthly for 6–8 months after vaccination. (**A**) Longitudinal 2nd-dose data of BNT162b2 (Pfizer) in comparison to mRNA-1273 (Moderna). (**B**) Longitudinal 3rd-dose data. (**C**) Longitudinal 3rd-dose data of individuals that received the same vaccine type for all three vaccine doses, in comparison to individuals that received two Pfizer doses and a Moderna booster dose. Data are shown as grouped box and whisker plots with error bars representing 5th–95th percentile of each population. Outliers outside of the 5th–95th percentile are shown as circular symbols above or below error bars. Graphs A and B data were analyzed using nonparametric Mann–Whitney test to evaluate mean rank between groups with a two-tailed *p*-value (*p* < 0.05) and 95% confidence interval (CI). Graph C data were analyzed using a nonparametric Kruskal–Wallis test to evaluate mean rank between groups using multiple comparisons (two-tailed *p* < 0.05 and 95% CI). Titers corresponding to percent neutralization ranges are described (see Figure 2 legend). * *p* < 0.05, ** *p* <0.005, *** *p* < 0.0005.

**Figure 4 vaccines-10-01459-f004:**
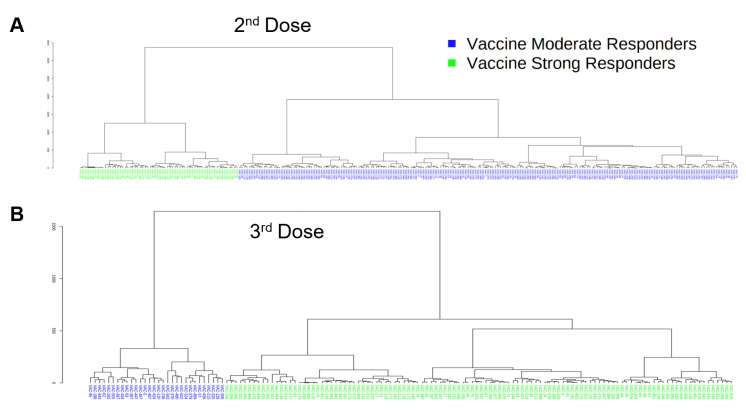
Second and third vaccine dose grouping analyses of unpaired longitudinal NAb data. (**A**,**B**) Unsupervised hierarchical clustering analysis performed using (**A**) 265 s dose subjects with percent neutralization data collected at pre-2nd dose, post-2nd dose, month 2, month 3, month 4, month 5, and months 6–8, and (**B**) 142 third-dose subjects with percent neutralization data collected at pre-3rd dose, post-3rd dose, month 2, month 3, month 4, month 5, and months 6–8. Dendrograms from both (**A**,**B**) show that the data are best classified as two groups such that within-group covariance is greater than between-group variance. In (**A**), 201 subjects were grouped as “vaccine moderate responders” while 64 subjects were classified as “vaccine strong responders” (VSR), while in (**B**), 30 subjects were classified as “vaccine moderate responders” (VMR) and 112 subjects were classified as “vaccine strong responders” (see (**C**,**D**)). (**C**,**D**) Line graphs showing percent neutralization by time following (**C**) 2nd dose and (**D**) 3rd dose between VSR and VMR. Data were grouped as indicated by unsupervised clustering (**A**,**B**). Error bars represent standard error. Significance determined by Mann–Whitney U test; FDR-controlled *q* values shown. * *q* < 0.05, *** *q* < 0.001. (**E**,**F**) PCA performed using percent neutralization values from (**E**) pre-2nd dose to months 6–8, and (**F**) pre-3rd dose to months 6–8. For both PCA score plots, subjects were classified as VSR or VMR via unsupervised clustering and significance analysis of measured differences in percent neutralization at each timepoint.

**Figure 5 vaccines-10-01459-f005:**
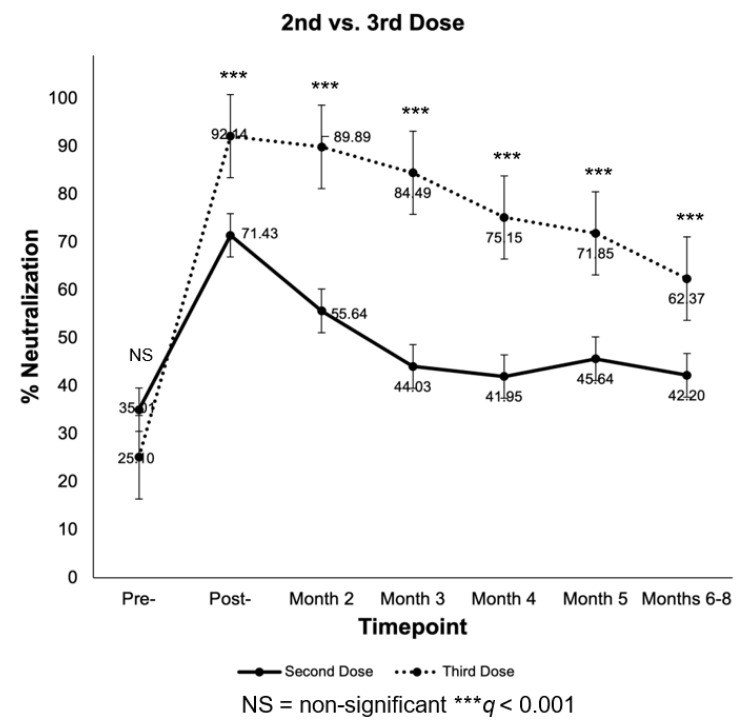
Line graph of measured percent neutralization between 2nd and 3rd vaccine dose recipients from pre-2nd and -3rd doses to months 6–8 follow-up. Error bars represent standard error. Significance determined by Mann–Whitney U test; FDR-controlled *q* values shown.

**Figure 6 vaccines-10-01459-f006:**
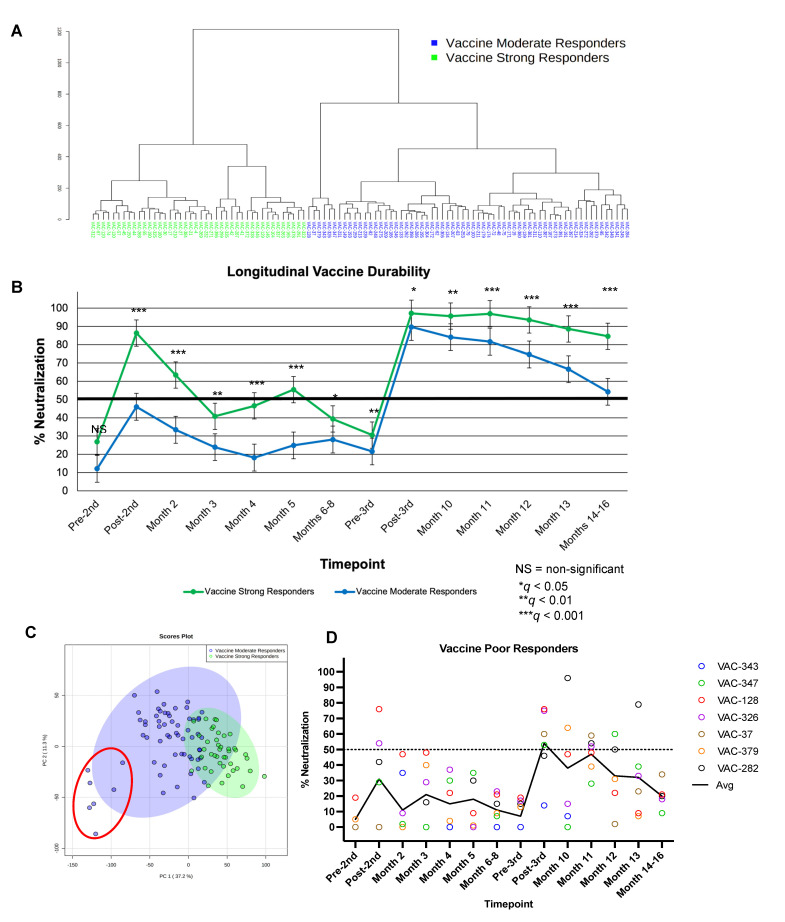
Grouping analyses of paired longitudinal NAb data. (**A**) Unsupervised hierarchical clustering analysis performed using 105 subjects with percent neutralization data collected at pre- and post-2nd dose, month 2, month 3, month 4, month 5, months 6–8, pre- and post-3rd dose, month 10, month 11, month 12, month 13, and months 14–16. Subjects were grouped as indicated by dendrogram and analysis of line graphs revealed the nature of two distinct groups (see (**B**)): VSRs (*n* = 42), VMRs (*n* = 63). (**B**) Line graph showing longitudinal vaccine durability (i.e., percent neutralization by time in response to 2nd and 3rd doses) between VSRs (*n* = 42) and VMRs (*n* = 63). Data were grouped as indicated by unsupervised clustering (**A**). Error bars represent standard error. Significance determined by Mann–Whitney *U* test; FDR-controlled *q* values shown. Referent line placed at 50% neutralization. (**C**) PCA performed using percent neutralization values from 105 subjects with longitudinal data from pre-2nd dose to months 14–16. Subjects were classified as VSR (green) or VMR (blue) via unsupervised clustering and significance analysis of measured differences in percent neutralization at each timepoint. A subgroup of the VMR population with poor neutralizing responses to mRNA vaccination, VPRs (*n* = 7), are shown outlined in red and loosely clustered within the VMR group (blue). (**D**) Paired longitudinal 2nd- and 3rd-dose data for VPR subjects identified in (**C**).

**Table 1 vaccines-10-01459-t001:** Population demographics of COVID-19 mRNA vaccine study participants.

Variable *	Total (N = 265) n (%)		Total (N = 142) n (%)	
	2nd Dose		3rd Dose	
	BNT162b2	mRNA-1273	BNT162b2	mRNA-1273
Sex				
Female	107 (40)	54 (20)	33 (23)	47 (33)
Male	65 (25)	39 (15)	30 (21)	32 (23)
Age				
(Median [range])	55 [20–82]	50 [19–73]	61 [26–81]	58 [20–79]
Female				
<65	87 (33)	47 (18)	21 (15)	34 (24)
≥65	20 (8)	7 (3)	12 (8)	12 (8)
Male				
<65	46 (17)	30 (11)	18 (13)	17 (12)
≥65	19 (7)	9 (3)	12 (9)	16 (11)

* Data shown as 2nd- and 3rd-dose population size (*N*). Sample sizes (*n*) are shown for sex and age subgroups with percentage of total population size (i.e., females < 65 years old that received a 2nd dose of BNT162b2 [*n* = 87] compose 33% of the total 2nd-dose population [*N* = 265]). All subjects included in analyses presented in this manuscript related to vaccine-induced NAbs were SARS-CoV-2 RT-PCR-negative at the time of enrollment and had no known breakthrough infections throughout the remainder of their enrollment. Subjects for which breakthrough infections did occur were included in vaccine-induced NAb data until the timepoint prior to infection, after which data were excluded. Such individuals are included in sample and population sizes above.

## Data Availability

All de-identified raw data corresponding to results presented in this manuscript are available upon reasonable request to the corresponding author.

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
