# Peer review of "Longitudinal Comparison of Neutralizing Antibody Responses to COVID-19 mRNA Vaccines after Second and Third Doses"

_vaccines, 2022, doi:10.3390/vaccines10091459_

Round 1
Reviewer 1 Report
Journal: Vaccines
Type of manuscript: Article
Manuscript ID: 1877290
Title: Longitudinal Comparison of Neutralizing Antibody Responses 2 to COVID-19 mRNA Vaccines after Second and Third Doses
Summary/General Reviewer Comments: In this longitudinal study, a novel rapid serology test was used to measure antibodies that block SARS-CoV-2 spike RBD-ACE2 receptor binding (“neutralizing” antibodies) in participants receiving second and third dose mRNA vaccines. As expected, the proportion of participants with high stable levels of antibodies increased with vaccine dosage. Participants with breakthrough infections were identified and high level neutralizing antibodies did not protect from clinical infection against the more recently emerging Omicron variant. This is a well written and methodologically constructed study that contributes to our understanding of COVID-19 mRNA vaccine efficacy using a new methodology that may better facilitate sero-surveillance.
Specific Reviewer Comments:
1) The validity of this study critically depends on the serological test performed. How well does the novel rapid-test accurately and reproducibly measure neutralizing antibodies, understanding that the results will be interpreted as protection from severe COVID-19 disease? Small blood volumes and single dilution antibody density measures might be more susceptible to reproducibility errors. Also, correlates between % neutralization and NAb titer appear to be inferred from a previous study (ref 17) instead of being performed coincident with this study.
2) The complexity of immune-mediated inactivation of SARS-CoV-2 is illustrated by the recent publication, CD8+ T Cells contribute to protection post vaccination (Liu J, Yu J, McMahan K, et al. CD8 T Cells Contribute to Vaccine Protection Against SARS-CoV-2 in Macaques. Science Immunology 0(0): eabq7647; 9 Aug 2022).
3) Methods, Line 88. “The purpose of this study was to quantify RBD-NAb titers ….”. As titers employ serial serum dilution and live virus to obtain quantitative information and are more representative of both antibody affinity and avidity and spectrum of neutralizing epitopes, perhaps a better term would be “to quantify RBD-NAb levels ….”.
4) Methods, Line 110. Individuals with PCR-confirmed natural infections were excluded from the study. However, many of the enrollees may have had undetected (untested) natural infections which could only be identified by serologic testing for non-spike protein antibodies.
5) Figure 1. Why do the authors interject race into the Figure? If they are concerned with excluding patients of color in their representation, they could simply present an outline of a hand and leave the color to the imagination of the viewer.
6) Table 1. Was race considered a study demographic variable?
7) Reference 17 should be: D. F. Lake, A. J. Roeder, E. Kaleta, P. Jasbi, K. Pfeffer, C. J. Koelbel, S. Periasamy, N. Kuzmina, A. Bukreyev, T. E. Grys, L. Wu, 624 J. R. Mills, K. McAulay, A. Seit-Nebi, S. Svarovsky. Development of a rapid point-of-care test that measures neutralizing antibodies to SARS-CoV-2. J Clin Virol 2021 Dec;145:105024. doi: 10.1016/j.jcv.2021.105024. Epub 2021 Nov 4.
Reviewer 2 Report
Overall, this is a well-written article, but the authors extend the analysis by exploring unplanned multiple subgroup comparisons beyond the original design and goals. In addition, some statements/conclusions regarding the protective effect of vaccination are not valid.
Some strengths of the study/article are:
- Estimation of quantitative titers of neutralizing antibodies in a lateral flow assay developed by authors.
- Antibodies titers were measured longitudinally before and after the 2nd and 3rd doses of the mRNA vaccine.
- Test neutralizing antibodies at multiple time points.
- Long-term follow-up of immune response after vaccination.
- Follow up of participants to detect Covid-19 cases and exclude them from the analysis.
- Assess of immune response following homologous and heterologous mRNA vaccination.
- Evidence of booster vaccination and sustained immune response after the 3rd dose, regardless of manufacturer.
- The study results can lead to further research and evaluation of vaccination strategies and immune responses.
Weaknesses and opportunities to improve the article are presented to the authors.
- I am concern about multi-stratified comparison of subgroups groups beyond the original study design and goals. In most cases, the sample size is not sufficient to draw reliable conclusions.
- I recommend focusing the analysis on the primary objective, rather than examining differences in immune responses across multiple subgroups.
- The Discussion section should be limited to discussing the implication of the results and conclusions, and should not repeat the results already presented in tables and graphs.
- Conclusions regarding prevention of infection, severe disease, hospitalization and lack of protection against Omicron infection are beyond the original purpose of the study and are uncertain as the study, sample size and follow-up strategy were not designed for this purpose.
-Based on the results, it would be interesting to propose new areas of research given the dynamics of Covid-19 infection and new variant-specific vaccines.
Author Response
Reviewer #2
Overall, this is a well-written article, but the authors extend the analysis by exploring unplanned multiple subgroup comparisons beyond the original design and goals. In addition, some statements/conclusions regarding the protective effect of vaccination are not valid.
Some strengths of the study/article are:
- Estimation of quantitative titers of neutralizing antibodies in a lateral flow assay developed by authors.
- Antibodies titers were measured longitudinally before and after the 2nd and 3rd doses of the mRNA vaccine.
- Test neutralizing antibodies at multiple time points.
- Long-term follow-up of immune response after vaccination.
- Follow up of participants to detect Covid-19 cases and exclude them from the analysis.
- Assess of immune response following homologous and heterologous mRNA vaccination.
- Evidence of booster vaccination and sustained immune response after the 3rd dose, regardless of manufacturer.
- The study results can lead to further research and evaluation of vaccination strategies and immune responses.
Weaknesses and opportunities to improve the article are presented to the authors.
- I am concern about multi-stratified comparison of subgroups groups beyond the original study design and goals. In most cases, the sample size is not sufficient to draw reliable conclusions.
- We performed post-hoc power analysis to assess the achieved beta (see lines 133-4) given alpha (0.05) sample size (N = 105) and effect size (R^2 = 0.3), and calculated the achieved power (1 – beta) to be 0.974 (see lines 362-5). These results have been summarized in Supplementary Figure S4. As our results are adequately powered (0.974) and are far above the conventional threshold for reliable power (0.80), we are indeed powered to make comparisons between stratified groups. Furthermore, it should be noted that estimates of study beta are related to type II error and therefore do not undermine our confidence (alpha) in observed differences across stratified subgroups.
- I recommend focusing the analysis on the primary objective, rather than examining differences in immune responses across multiple subgroups.
- We view the primary objective in the manuscript to be Figure 2, illustrating the importance and durability of a third vaccine dose. However, it was important to dive more deeply into our observations to examine the heterogeneity of NAb responses to COVID-19 mRNA vaccines, and to determine if any of the vaccine strong responders, moderate responders, or poor responders were represented by a particular demographic.
- The Discussion section should be limited to discussing the implication of the results and conclusions, and should not repeat the results already presented in tables and graphs.
- We agree with the reviewer and have removed repetitious results from the discussion section of the main text. Original draft line numbers 434-436,445-448, 462-466, and 502-504 were removed, reflected by “tracked changes” in Microsoft Word.
- Conclusions regarding prevention of infection, severe disease, hospitalization and lack of protection against Omicron infection are beyond the original purpose of the study and are uncertain as the study, sample size and follow-up strategy were not designed for this purpose.
- We agree with the reviewer. During the last few months of our longitudinal study, Omicron became the dominant variant among the population and breakthrough infections increased in frequency despite participants’ high levels of NAb to their Wuhan-based vaccine. We believe the Omicron breakthrough figure enriches the narrative of the outcome of our study. Nonetheless, we have moved Figure 7 into the Supplementary Appendix (now Supplementary Figure S5).
- Based on the results, it would be interesting to propose new areas of research given the dynamics of Covid-19 infection and new variant-specific vaccines.
- Thank you for the suggestion. We have added to the discussion the possibility of using the rapid test to monitor individual responses to new variant-specific vaccines.

Round 2
Reviewer 2 Report
Thank you for the revised version of the manuscript. I noticed the changes and clarification of the points raised.